Citation: *Molecular Systems Biology* 9:686
www.molecularsystemsbiology.com

# Chromosome segregation by the *Escherichia coli* Min system

Barbara Di Ventura[1,5,*], Benoît Knecht[2], Helena Andreas[3], William J Godinez[4], Miriam Fritsche[2], Karl Rohr[4], Walter Nickel[3], Dieter W Heermann[2] and Victor Sourjik[1,*]

[1] Zentrum für Molekulare Biologie der Universität Heidelberg, DKFZ-ZMBH Alliance, Heidelberg, Germany, [2] Institute for Theoretical Physics, University of Heidelberg, Heidelberg, Germany, [3] Heidelberg University Biochemistry Center, University of Heidelberg, Heidelberg, Germany and [4] Department of Bioinformatics and Functional Genomics, Biomedical Computer Vision Group, Institute for Pharmacy and Molecular Biotechnology (IPMB), BioQuant and DKFZ, University of Heidelberg, Heidelberg, Germany
[5] Present address: Department of Bioinformatics and Functional Genomics, Synthetic Biology Group, Institute for Pharmacy and Biotechnology (IPMB) and BioQuant, University of Heidelberg, Heidelberg, Germany

* Corresponding authors. B Di Ventura, BioQuant, University of Heidelberg, Im Neuenheimer Feld 267, Heidelberg 69120, Germany. Tel.: + 49 6221 54 51283; Fax: + 49 6221 54 51488; E-mail: barbara.diventura@bioquant.uni-heidelberg.de or V Sourjik, Zentrum für Molekulare Biologie der Universität Heidelberg, DKFZ-ZMBH Alliance, Heidelberg, Germany. Tel.: + 49 6221 54 6958; Fax: + 49 6221 54 5892; E-mail: v.sourjik@zmbh.uni-heidelberg.de

**The mechanisms underlying chromosome segregation in prokaryotes remain a subject of debate and no unifying view has yet emerged. Given that the initial disentanglement of duplicated chromosomes could be achieved by purely entropic forces, even the requirement of an active prokaryotic segregation machinery has been questioned. Using computer simulations, we show that entropic forces alone are not sufficient to achieve and maintain full separation of chromosomes. This is, however, possible by assuming repeated binding of chromosomes along a gradient of membrane-associated tethering sites toward the poles. We propose that, in *Escherichia coli*, such a gradient of membrane tethering sites may be provided by the oscillatory Min system, otherwise known for its role in selecting the cell division site. Consistent with this hypothesis, we demonstrate that MinD binds to DNA and tethers it to the membrane in an ATP-dependent manner. Taken together, our combined theoretical and experimental results suggest the existence of a novel mechanism of chromosome segregation based on the Min system, further highlighting the importance of active segregation of chromosomes in prokaryotic cell biology.**
*Molecular Systems Biology* **9**: 686; published online 10 September 2013; doi:10.1038/msb.2013.44
*Subject Categories:* simulation and data analysis; cell cycle
*Keywords:* computer simulations; chromosome segregation; DNA binding; MinD; Min system

## Introduction

When cells divide, their genetic content has to be faithfully copied and equally distributed to the progeny. Several processes, therefore, exist and cooperate to carry out the delicate task of cell division, such as DNA replication, segregation, correct positioning of the division site, and cytokinesis itself. For prokaryotes, much is known about how the DNA is replicated (Scholefield *et al*, 2011; Badrinarayanan *et al*, 2012), how cells define their middle in a precise way (Thanbichler and Shapiro, 2008; de Boer, 2010; Lutkenhaus, 2012), and how cytokinesis is carried out (de Boer, 2010; Erickson *et al*, 2010). The mechanisms of chromosome segregation have, on the other hand, remained largely enigmatic (Pogliano *et al*, 2003; Reyes-Lamothe *et al*, 2012), and different models propose segregation to be either passive or active. In the model of purely passive segregation, the forces that separate sister chromosomes are internal to the chromosomes themselves and are generated by repulsion of two self-avoiding polymers in a rod-shaped geometry to maximize their conformational entropy (Jun and Mulder, 2006; Jun and

Wright, 2010). Partitioning forces may also arise from a number of other processes (Toro and Shapiro, 2010) such as the interplay between the organization of the nucleoid and replication (Sawitzke and Austin, 2001) or co-transcriptional translation and translocation of membrane proteins (Woldringh, 2002). In the models of active segregation, external forces produced by specialized proteins use energy to move duplicated chromosomes each into one daughter cell, more closely resembling the function of the eukaryotic mitotic apparatus. The presence of a dedicated segregation machinery has been recently shown in *Caulobacter crescentus* (Ptacin *et al*, 2010; Schofield *et al*, 2010; Shebelut *et al*, 2010) and in *Vibrio cholerae* (Fogel and Waldor, 2006). This machinery was proposed to rely on the force generated by depolymerization of oligomers that are formed by the cytoplasmic DNA-binding ATPase ParA. Similar Par systems are also involved in the segregation of some low-copy-number plasmids in bacteria (Ringgaard *et al*, 2009; Gerdes *et al*, 2010). However, *E. coli* and many other bacteria lack a chromosomal Par system, suggesting that the ParA-dependent segregation mechanism is not universal.

The closest homolog of ParA in *E. coli*, the ATPase MinD, is part of the Min system that has a well-established function in restricting the division plane to mid-cell (Lutkenhaus, 2007, 2012). MinD has an additional C-terminal amphipathic helix that allows it to form membrane-associated, ATP-dependent dynamic filaments (Hu *et al*, 2002; Szeto *et al*, 2002; Hu and Lutkenhaus, 2003; Szeto *et al*, 2003; Zhou and Lutkenhaus, 2003), which exhibit periodic pole-to-pole oscillations in the cell. Min oscillations arise from the interplay between the ATP-dependent membrane association and subsequent oligomerization of MinD, and MinE-stimulated local release of MinD from the membrane upon ATP hydrolysis (Hu *et al*, 2002; Lackner *et al*, 2003; Kruse *et al*, 2007; Loose *et al*, 2008). These oscillations create an intracellular gradient of the complex between MinD and the cell-division inhibitor MinC (Raskin and de Boer, 1999), with a minimum at mid-cell and maxima at the poles.

In this study, we use numerical computer simulations to demonstrate how a gradient of DNA binding sites at the cell membrane can act as a Brownian ratchet to bias the movement of chromosomes from mid-cell toward the poles, completing and maintaining chromosome segregation initially achieved by purely entropic repulsion forces. We further propose that such a gradient can be provided by the Min system, demonstrating that MinD can bind to DNA and tether it to the membrane in an ATP-dependent manner. These results suggest a novel mechanism of active chromosome segregation that might be common among bacteria.

## Results

### Polar gradients of DNA binding sites at the membrane can enhance entropy-driven segregation of chromosomes

Previous computer models of chromosome segregation showed that entropic repulsion might be sufficient to promote the initial disentanglement of two self-avoiding ring polymers representing the duplicated DNA (Jun and Mulder, 2006; Jun and Wright, 2010). Yet because the entropic forces are expected to drop sharply after the initial unmixing of such polymers, entropy alone is unlikely to ensure clearance of chromosomes away from mid-cell, and therefore additional mechanism(s) must exist in *E. coli* to complete chromosome segregation.

We hypothesized that one such mechanism may be provided by polar gradients of DNA tethering sites at the membrane, whereby repeated binding and unbinding of chromosomes to these sites would prevent backward movement of the DNA toward mid-cell, effectively biasing its random diffusion toward the poles and thus resulting in a Brownian ratchet-type mechanism of segregation. To test this hypothesis, we simulated the dynamics of two self-avoiding ring polymers ( = chromosomes) confined in a volume with an aspect ratio of 1:8 corresponding to that of an *E. coli* cell (Supplementary Figure S1A). In these simulations, entropic repulsion is represented by the excluded volume interactions between the two ring polymers, as well as between the segments of one polymer, meaning that two segments cannot cross or overlap (Bohn and Heermann, 2010; Jun and Wright, 2010; Fritsche *et al*, 2012). We further considered membrane tethering of

chromosomal segments, with either a homogenous or polar gradients distribution of such sites. A typical simulation starts with the two polymers being mixed (Supplementary Figure S1A) and is run by stepwise displacement of polymer segments using a Monte-Carlo method until the centers of mass of both polymers have reached their steady-state position. As shown in Figure 1A, polar gradients of DNA tethering sites on the membrane lead to a more pronounced separation of the two polymers than purely entropic repulsion. Similar improvement in segregation was obtained with static or dynamic gradients, whereby in the latter case the gradient was allowed to periodically oscillate (Supplementary Figure S1B and C). Notably, a uniform distribution of tethering sites does not improve and even slows segregation down. Decreasing the steepness of the gradient makes segregation less efficient but does not fully impair it (Supplementary Figure S1D). In contrast, increasing the dwell time (and therefore effective affinity) of DNA segments at tethering sites by 10-fold reduces the efficiency of segregation below that accomplished by the entropic repulsion alone (Supplementary Figure S1E). Thus, efficient segregation requires DNA tethering to be relatively weak and transient.

The benefit of the proposed mechanism for chromosome segregation is even more evident when comparing the polymer density profiles over time in simulations without (Figure 1B) and with oscillating gradient of tethering sites (Figure 1C). This confirms that while the entropic repulsion of the nucleoids is sufficient to initially push chromosomes apart, it subsequently becomes too weak to achieve full segregation away from mid-cell. At this point, the action of the Brownian ratchet that is mediated by a gradient of binding sites becomes important. Furthermore, comparing the distribution of the center of polymer mass upon equilibration in multiple simulations reveals that only with a gradient of tethering sites the center of mass is positioned with high precision (Figure 1D). Importantly, such a gradient is able to efficiently segregate polymers even independent of the entropic forces (Figure 1E), and while the entropic repulsive force drops rapidly with increasing distance between the centers of mass of the two polymers, the gradient of tethering sites can maintain the effective repulsion at larger distances where the entropic contribution becomes negligible (Figure 1F). Notably, in our model chromosome segregation is primarily generated by tethering sites that are distributed along the lateral membrane. As a consequence, our computer simulations neither show any pronounced extension of the polymers toward the cell poles nor require such extension for segregation, which is consistent with the observed nucleoid morphology in *E. coli* cells.

### Identification of MinD as a candidate tethering protein

In principle, the proposed mechanism of chromosome segregation can be mediated by any protein (or protein complex) that forms polar gradients at the membrane and binds DNA. Since MinD is known to form dynamic polar gradients in *E. coli* and given its homology to ParA, we decided to test whether it could also bind DNA. Indeed, we found that incubation with MinD alters the electrophoretic mobility of

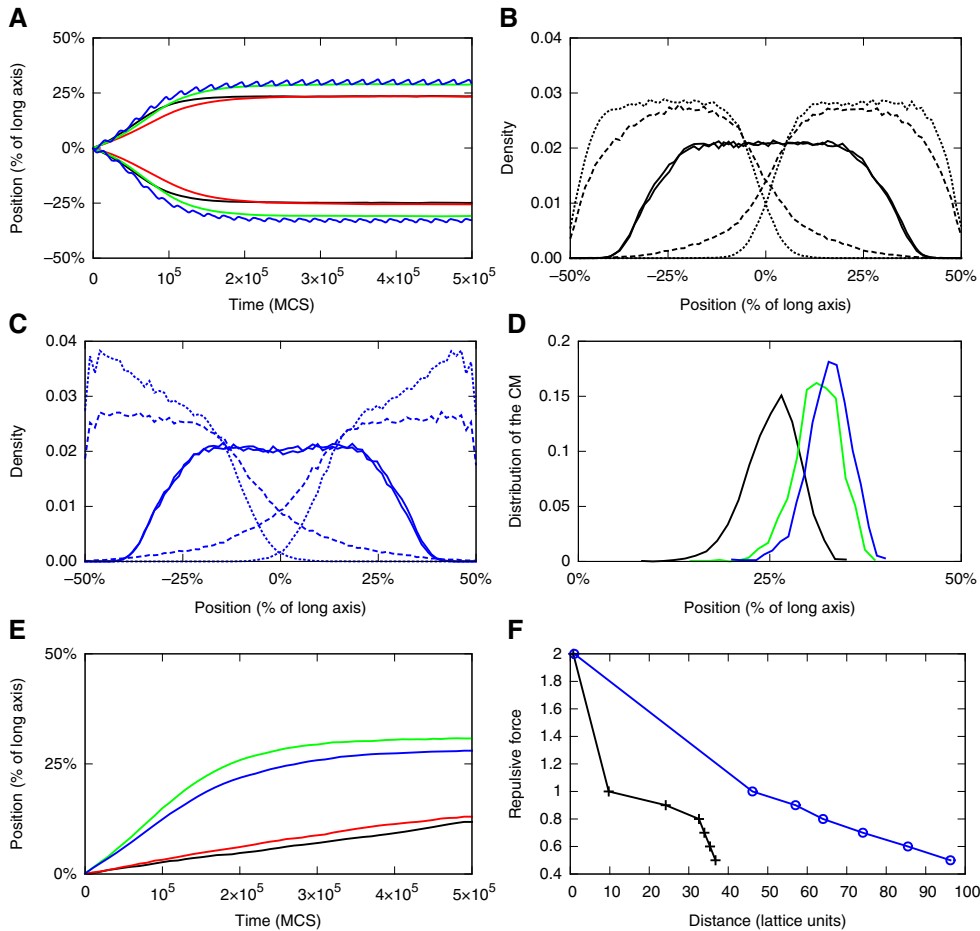

**Figure 1** Simulated effects of a gradient of membrane tethering sites on chromosome segregation. Monte-Carlo simulations were performed in elongated rectangular parallelepiped geometry of aspect ratio 1:8 and with $80 \times 10 \times 10$ lattice size, treating chromosomes as two self-avoiding ring polymers. Chromosome dynamics was simulated without tethering (black), with membrane tethering sites distributed uniformly (red), or with a static (green) or a dynamic (blue) gradient toward the poles. (**A**) Position of the center of mass (CM) of each polymer relative to the long cell axis as a function of time, measured in Monte-Carlo steps (MCS). (**B**, **C**) Density profiles for each polymer along the long cell axis in simulations without tethering (**B**) or with an oscillating gradient of tethering sites (**C**) at 0 (continuous line), 105 (dashed line), and $5 \times 10^5$ (dotted line) MCS. (**D**) Distribution of CM of the right polymer along the long cell axis, obtained for 2048 simulations without tethering, or with a static or an oscillating gradient of tethering sites. (**E**) Same as (A) but simulated without entropic repulsion between polymers. Only position of the right polymer along the long cell axis is shown. (**F**) Repulsive force between two polymers as a function of the distance between their CMs, for the model without tethering or with an oscillating gradient. The repulsive force was computed by applying a given attractive force between the polymers and measuring the distance at which they equilibrate.

DNA fragments in the electrophoretic mobility shift assay (EMSA), retaining a large fraction of the 155 bp double-stranded DNA probe in the well (Figure 2A). This shift largely in the upper portion of the gel indicated formation of high molecular weight (HMW) nucleoprotein filaments. The inter-action was not sequence specific, being observed to a similar extent for DNA fragments that correspond to the P1 promoter of the *E. coli minB* operon and to the unrelated hybrid *pTrc* promoter. Confirming that HMW nucleoprotein complexes result from binding of multiple MinD proteins to the same DNA molecule rather than from MinD aggregation, smaller MinD-bound DNA fragments migrated into the gel as distinct bands (Figure 2B and C). The HMW MinD–DNA complexes appeared to be inhibited by ADP (Figure 2A and B), since they were much more pronounced in the presence of ATP or in the absence of any nucleotide added to the reaction. We further showed that in order for MinD to bind, the DNA fragment has to be longer than 10 bp (Figure 2C). The distinct band observed

for the 20–30 bp DNA fragments (Figure 2B and C) thus likely corresponds to a DNA-bound dimer of MinD. In contrast, a variable number of MinD proteins can bind to longer DNA fragments, resulting in the formation of a smear due to the multiple species present (Figure 2A).

## Conserved arginine 219 is involved in MinD binding to DNA

We next characterized the effect of several mutations that are known to affect either DNA binding by the ParA family of ATPases or MinD activity (Figure 2D and E; Supplementary Figure S2; Supplementary Table S2). Indeed, aspartate replace-ment of arginine 219 (R219D) had a strong negative effect on DNA binding (Figure 2E; Supplementary Figure S2A and B). This residue corresponds to positively charged residues that are important for the non-sequence specific DNA binding of

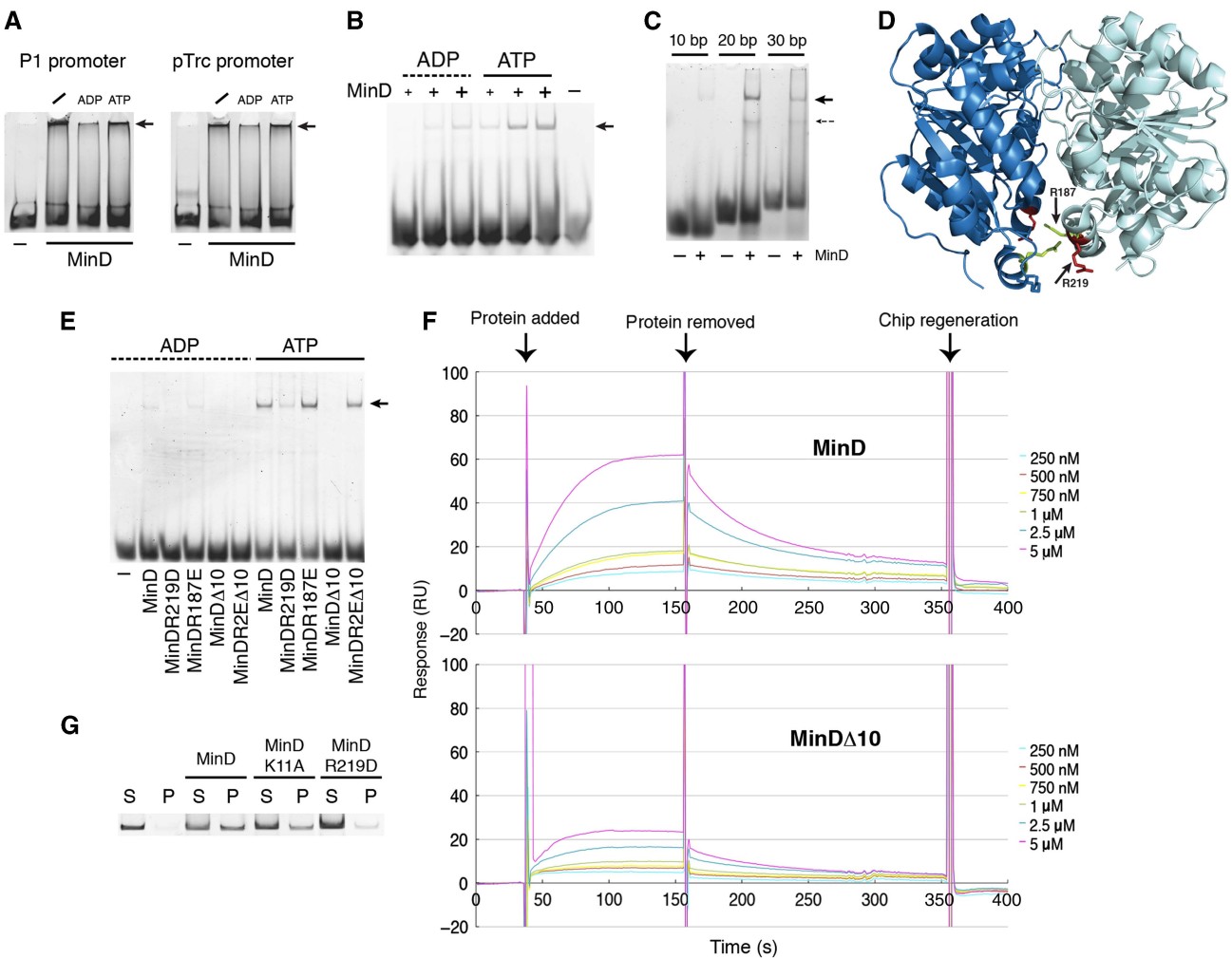

**Figure 2** Characterization of DNA binding by MinD. (**A–C**, **E**) Electrophoretic mobility shift assays (EMSAs) of dsDNA fragments (200 fmol) labeled by 5′-hexachloro-6-carboxy-fluoresceine (HEX) incubated in the presence of 1 mM ATP or ADP with or without (minus) MinD. Samples were separated on 6% native polyacrylamide (PA) gel. Arrows indicate nucleoprotein complexes. (**A**) 155 bp DNA fragments corresponding either to the P1 promoter of the *minB* operon or to the synthetic pTrc promoter were incubated with 5.5 µM of wild-type MinD. Slash indicates samples without nucleotides. (**B**) 28 bp fragment of the pTrc promoter was incubated with varying amounts of MinD (0.6, 1.8, and 4 µM), indicated by plus symbols of increasing size. (**C**) P1 promoter fragments of the indicated size were incubated with 5.5 µM of MinD and ATP. (**D**) Schematic view of the MinD structure (PBD accession number 3Q9L), oriented above the membrane (Wu *et al*, 2011). Two residues tested for their effects on DNA binding are highlighted. (**E**) 28-bp pTrc probe was incubated with 2.5 µM of wild-type or indicated mutant MinD proteins (see text for details). (**F**) Surface plasmon resonance analysis of DNA binding by wild-type and C-terminally truncated MinD (MinD$^{\Delta10}$). See Supplementary information for a detailed explanation of $k_{on}$ and $k_{off}$ estimation. (**G**) MinD-DNA co-sedimentation assay. Wild-type or mutant MinD proteins (1 µM) were incubated with HEX-labeled P1 promoter (200 fmol) and ATP (1 mM) and pelleted by centrifugation at 21 000 *g* for 30 min. Supernatant (S) and pellet (P) fractions were run on 10% PA gel and HEX-DNA was visualized. A negative control without protein is also shown.

ParA-family members, arginine 218 in Soj and lysine 340 in SopA (Hester and Lutkenhaus, 2007; Castaing *et al*, 2008). However, replacement of arginine 187, which aligns with another important DNA-binding residue of Soj, arginine 189 (Hayes and Barilla, 2006), had no effect (Figure 2E). In an attempt to find other residues that could be involved in DNA binding, we tested two arginines at positions 251 and 254, but found that single (MinD$^{R251E}$ and MinD$^{R254E}$) or double (MinD$^{R2E}$) mutation of these residues to glutamates had no effect on DNA binding (Supplementary Table S2). We also mutated other, positively charged residues lying at the core of the MinD dimer, but found no effect (Supplementary Table S2). To our surprise, DNA binding was nearly abolished by truncation of the last 10 C-terminal residues that form the

amphipathic helix, normally responsible for membrane association of MinD (MinD$^{\Delta10}$; Figure 2E and Supplementary Figure S2A). Similar results were obtained using surface plasmon resonance (SPR) assays (Figure 2F), which also showed that DNA binding and dissociation of MinD *in vitro* occurs at the time scale of tens of seconds, with an apparent dissociation constant of ~0.6 µM. Nevertheless, the amphipathic helix is unlikely to be directly involved in DNA binding, since the binding could be restored by introducing the R2E mutation in the context of the truncated MinD (MinD$^{R2E\Delta10}$; Figure 2E). Moreover, the ability of this mutant to bind to DNA strongly suggests that MinD non-sequence specific DNA binding is not due to its being positively charged at the C-terminus, since MinD$^{R2E\Delta10}$ not only lacks the positively

charged C-terminal helix but even contains two more negatively charged residues compared with wild-type MinD. We propose that the C-terminal helix modulates DNA binding of MinD via a conformational change, which may be mimicked by mutating arginines 251 and 254 to glutamates in the truncated MinD.

## MinD forms HMW complexes with DNA

Formation of the HMW MinD complexes with DNA was further confirmed by sedimentation analysis of the labeled DNA probe with and without MinD. Indeed, DNA was found in the pellet in the presence of wild-type MinD and ATP (Figure 2G). The sedimentation of DNA was strongly reduced in the presence of mutant MinD$^{R219D}$ (Figure 2G). Interestingly, substantial sedimentation of DNA was observed for MinD$^{K11A}$ mutant that is not able to dimerize (Zhou *et al*, 2005) but is still able to bind ATP (Okuno *et al*, 2010), indicating that binding of multiple monomers to the same DNA fragment may be sufficient to form HMW nucleoprotein complexes. MinD$^{K11A}$–DNA complexes were also detected in EMSAs (Supplementary Figure S2B).

## MinD can tether DNA to the membrane in an ATP-dependent manner

To test whether MinD is able to tether the DNA to the membrane, we used a flotation assay (Weber *et al*, 1998) in which a mixture of protein, DNA, and liposomes was separated by ultracentrifugation in a density gradient. Under our experimental conditions, liposomes and liposome-associated molecules move to the top of the gradient, whereas

proteins and DNA that are not bound to liposomes remain at the bottom (Figure 3A). The ultracentrifugation is carried out for 4 h allowing the material in the gradient to reach equilibrium (Steringer *et al*, 2012). As expected, when subject to separation individually, liposomes were primarily found in the top fraction 1 (Figure 3B), whereas DNA was found in the bottom fractions 3 and 4 (Supplementary Figure S3A). When mixed together, DNA and MinD were found in fractions 3 and 4 (Supplementary Figure S3B). In the presence of MinD and ATP, liposomes were also found in fraction 2 (Figure 3C; Supplementary Figure S3D), indicating the formation of complexes between MinD and liposomes that are substantially heavier than free liposomes and might correspond to the previously observed MinD-induced membrane tubules (Hu *et al*, 2002). In the presence of MinD, DNA also became enriched in fraction 2 in an ATP-dependent manner (Figure 3D; Supplementary Figure S3D), suggesting that MinD oligomers are able to recruit DNA to the membrane. Certain liposome-dependent enrichment of DNA in fraction 1 was observed even in the absence of MinD (Supplementary Figure S3C), presumably due to a non-specific binding. Nevertheless, the MinD- and ATP-dependent recruitment of DNA to liposomes in fraction 2 was much more efficient, confirming its specificity (Figure 3E; Supplementary Figure S3C and D).

In the flotation assay, MinD also localized to the lighter liposome fraction 1, in the presence of either ATP or ADP (Supplementary Figure S3D), indicating that the assay is sensitive enough to detect the weak membrane binding of monomeric MinD (Szeto *et al*, 2003). However, the higher ratio of DNA to MinD in fraction 2 suggests that oligomeric ATP-bound MinD is much more potent in recruiting DNA to the liposomes (Figure 3F). Consistent with this explanation, monomeric MinD$^{K11A}$ that, as expected, localized to

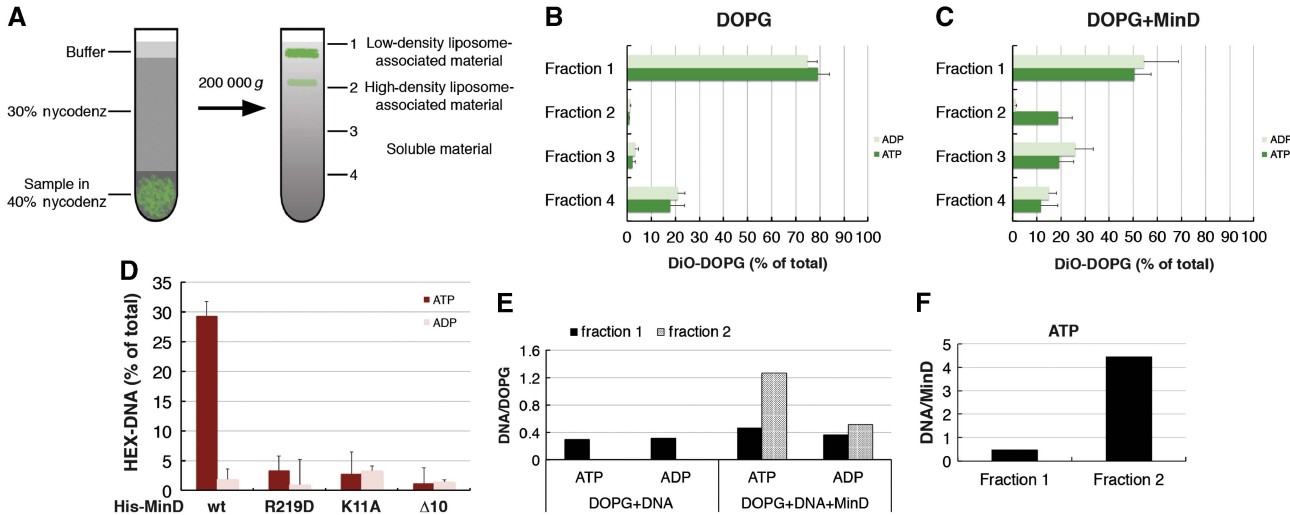

**Figure 3** MinD tethers DNA to liposomes in an ATP-dependent manner. (**A**) Schematic overview of the flotation experiment with DOPG (1,2-dioleoyl-*sn*-glycero-3-phospho-(1′-*rac*-glycerol)) liposomes, fluorescently labeled with 0.1% DiO (3,3′-dioctadecyloxacarbocyanine perchlorate). Upon ultracentrifugation in the Nycodenz density gradient, free liposomes move to the top of the gradient (fraction 1), whereas soluble material remains at the bottom (fractions 3 and 4). Heavier (high density) liposome–protein complexes are found in fraction 2. Relative distribution of DiO-labeled DOPG liposomes (400 μg/ml) to individual fractions in the flotation assay in the absence (**B**) or presence (**C**) of MinD (1.3 μM). (**D**) Flotation assays with MinD mutants, performed as in (C) including HEX-labeled DNA (155 bp, P1 promoter, 200 fmol) to the reactions. Only fraction 2 is shown here; other fractions are shown in Supplementary Figure S3D–G. 1 mM ADP or ATP was added to the reaction, as indicated. The *x* axes represent the amount of material that floted to each fraction as the percentage of the total material recovered from the gradient. (**E**) Bar plot showing the ratio between the amounts of DNA and DOPG liposomes found in fractions 1 and 2. For the DOPG + DNA case, only fraction 1 is shown, as liposomes do not flot to fraction 2 in the absence of MinD. (**F**) Bar plot showing the ratio between the amounts of DNA and MinD found in fractions 1 and 2.

fraction 1 but was not able to shift liposomes to fraction 2, led only to a slight enrichment of DNA in fraction 1 (Figure 3D; Supplementary Figure S3F). MinD$^{\Delta 10}$ was completely deficient in liposome binding (Figure 3D; Supplementary Figure S3G).

The DNA-binding mutant MinD$^{R219D}$ was largely impaired in tethering DNA to the membrane (Figure 3D). Yet this result cannot be unambiguously assigned to the impairment of MinD$^{R219D}$ DNA binding, since the R219D mutation also apparently affects binding of MinD to the membrane. MinD$^{R219D}$ could not efficiently shift the liposomes to fraction 2, although it was found in fraction 1 (Supplementary Figure S3E). Weaker binding of MinD$^{R219D}$ to the membrane was further confirmed using a liposome sedimentation assay,

with significantly smaller amount of liposome-bound MinD$^{R219D}$ in the pellet compared with the wild-type MinD (Figure 4A; Supplementary Figure S4). Considering the location of arginine 219 on the surface of MinD that faces toward the membrane (Wu *et al*, 2011), it is perhaps not surprising that this mutation affects MinD interaction with the membrane. Yet, to our knowledge, this is the first mutation mapped outside of the C-terminal helix that specifically affects binding of MinD to the membrane. Moreover, the involvement of residues outside of the C-terminal amphipathic helix in membrane binding was confirmed by weaker liposome-mediated sedimentation of MinD carrying R251E and R254E mutations (Figure 4B).

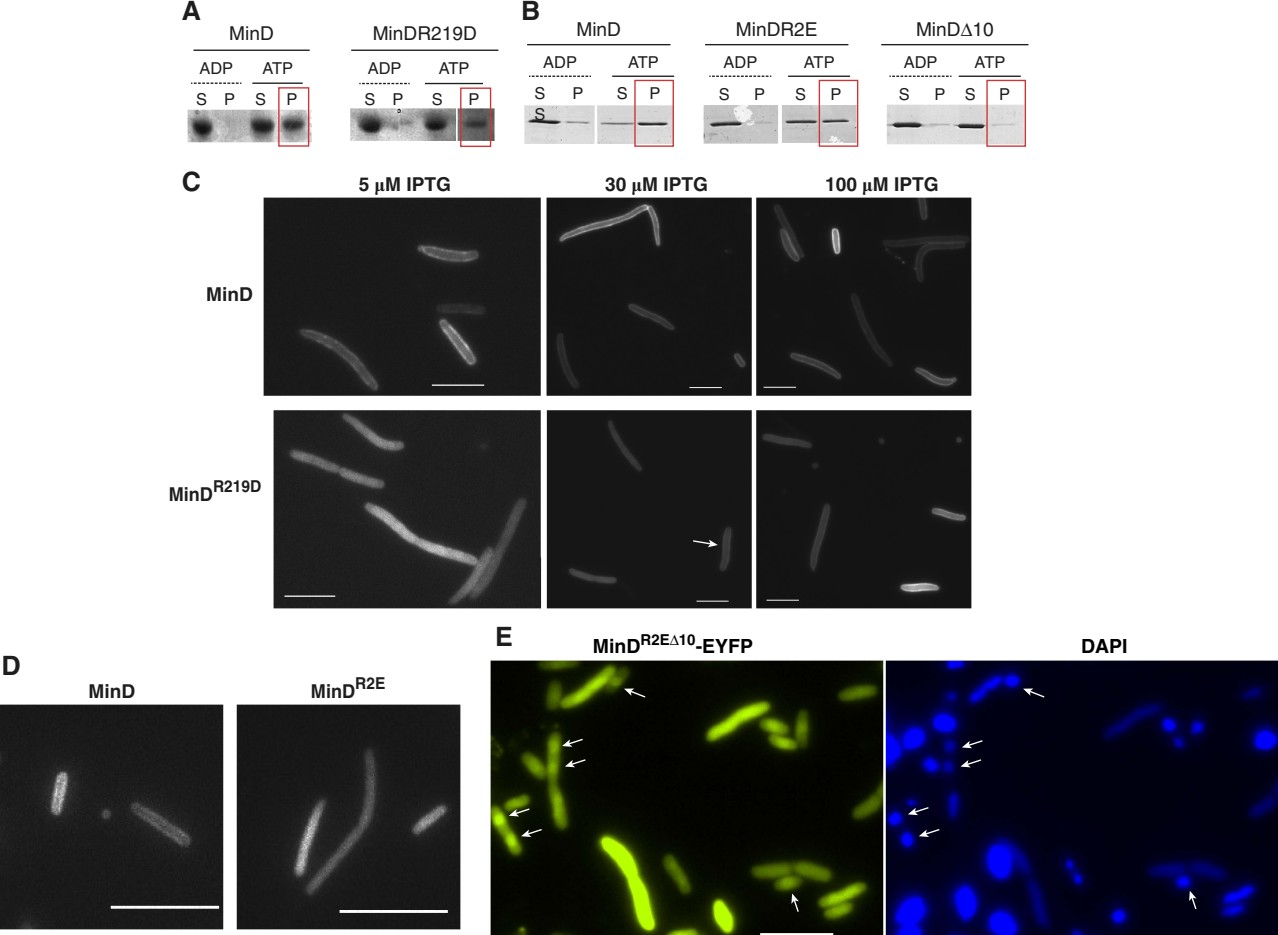

**Figure 4** DNA and membrane binding of MinD are interconnected. (**A**, **B**) Co-sedimentation assay of MinD (wild type or mutant, as indicated) with liposomes. 1-$\mu$M protein was incubated with liposomes (320 $\mu$g/ml) and either ADP or ATP (1 mM). Reactions were incubated for 10 min at RT and then centrifuged at 21 000 *g* for 30 min. Pellets were resuspended in reaction buffer (see Supplementary information) and supernatants and pellets were analyzed by SDS–PAGE, visualized by Coomassie staining. Since the pellet fraction represents material that is bound to the liposomes, the presence of protein in this fraction is indicative of binding. A red box around the pellets in the ATP case is drawn to help comparing the liposome-binding capacity of the various MinD proteins. The white space indicates deletion of the lane containing the protein marker. (**C**) Representative images of $\Delta minB$ cells expressing either wild-type (upper panel) or mutant MinD$^{R219D}$ (lower panel). Cells were grown in LB medium at 34$^\circ$C (220 r.p.m.) until early exponential phase (OD$_{600} \approx 0.2$), then IPTG (at the indicated levels) was added to induce expression of fluorescently labeled MinD. Induction was carried out for 1 h and then cells were harvested and prepared for microscopy as described in Supplementary information. The white arrow points to a cell with membrane-bound MinD$^{R219D}$. (**D**) Same as in (C), but for another MinD mutant, in which the two ariginines at positions 251 and 255 were substituted by glutamic acids (MinD$^{R2E}$). Induction was as in (C). (**E**) Representative images of MG1655 cells expressing MinD$^{R2E\Delta 10}$-EYFP. Cells were grown as in (C) but at 37$^\circ$C and in the presence of 100 $\mu$M IPTG for 3 h. To visualize the nucleoid, cells were incubated with 10 $\mu$l of DAPI solution (0.5 $\mu$g/ml in 50% glycerol) on the agarose pads for 5 min. White arrows point to some of the cells that show an enrichment of the EYFP fluorescence in the DAPI-stained chromosomal regions. Notably, there is no bleed-through of the DAPI signal into the EYFP channel, as evidenced by several cells showing an extremely bright DAPI signal that are not visible in the EYFP channel. (**C–E**) Scale bar, 5 $\mu$m.

We further investigated the behavior of these mutants *in vivo*. When fused to a yellow fluorescent protein (YFP), MinD$^{R219D}$ was able to bind to the membrane (Figure 4C) and to support oscillations when expressed together with MinE (Supplementary Movies 1 and 3). However, both membrane binding and oscillations required higher expression levels than for wild-type MinD, consistent with the lower affinity of this mutant for the membrane. Moreover, MinD$^{R219D}$ apparently could not support MinC oscillations even when using induction levels that led to MinDE oscillations (Supplementary Movies 2, 4 and 5), indicating that the R219D mutation might also directly affect MinD binding to MinC. The negative effect of R251E and R254E mutations on membrane binding of MinD could be also confirmed *in vivo* (Figure 4D).

## The cytoplasmic mutant MinD$^{R2E\Delta10}$ is enriched on the nucleoid *in vivo*

As previously mentioned, the R251E and R254E mutations in the context of the truncated MinD$^{\Delta10}$ can restore DNA binding. As this mutant (MinD$^{R2E\Delta10}$) is cytoplasmic, we reckoned that it might show co-localization with the nucleoid in *E. coli* cells that is otherwise obscured by binding of MinD to the membrane. We therefore constructed a fusion of MinD$^{R2E\Delta10}$ to EYFP and analyzed its *in vivo* localization. Albeit not in all cells, MinD$^{R2E\Delta10}$-EYFP clearly showed enrichment over the nucleoid area, as ascertained by imaging both the EYFP and the DAPI channels (Figure 4E). The co-localization of MinD$^{R2E\Delta10}$-EYFP with DNA was only partial, supporting the prediction of our computer simulations that MinD-DNA binding has to be intrinsically weak.

## Absence of MinD leads to an increase in anucleate cells

If MinD is involved in chromosome segregation, then one might expect to observe segregation defects in cells lacking MinD. To distinguish specific MinD-dependent segregation defects from those caused by the asymmetric cell division in the absence of the functional Min system, we compared the distribution of DAPI-stained DNA in a population of Δ*minB* cells lacking the entire Min system (MinC, MinD and MinE) to that in Δ*minC* cells lacking only the inhibitor of cell division MinC. Because of aberrant division near the cell poles in the absence of MinC, both these strains produce anucleate mini-cells, the hallmark of the *min* mutants. Nevertheless, cells lacking only MinC show visibly better separation of the nucleoids compared with those lacking all Min proteins (Figure 5A; Supplementary Figure S5), supporting our idea that MinD has a role in chromosome segregation that is independent of its function in placement of the cell-division site. Moreover, we found an increase in anucleate cells in the Δ*minB* strain compared with the Δ*minC* strain (Figure 5B) when excluding mini-cells from our analysis. Even assuming that the determined fraction of anucleate cells is overestimated because DAPI staining of DNA did not occur in all cells, the observed specific difference between the Δ*minB* and Δ*minC* strains strongly suggests that the defect in the MinD-dependent partitioning can lead to the loss of chromosomes also in non-mini cells.

## Mutation of arginine 219 in MinD causes chromosome segregation defects *in vivo*

An even more direct proof of the involvement of MinD in chromosome segregation would be given by a MinD mutant that is not able to bind DNA, but can still oscillate and thus mediate the cell-size control. However, because of the observed interrelation between the two activities of MinD, we could not effectively decouple the defects in DNA and membrane binding. Indeed, consistent with its lower affinity for the membrane and for MinC, MinD$^{R219D}$ could not fully complement the mini-cell phenotype of Δ*minB* strain (deletion of all three Min proteins) even when co-expressed with MinC and MinE. Nonetheless, we reasoned that the effect of the R219D mutation on DNA segregation by MinD might become apparent when comparing Δ*minB* cells expressing either MinD/MinE or MinD$^{R219D}$/MinE. In this experiment, we also expressed a basal level of MinC, since this gave us a similar distribution of cell sizes in both strains, decreasing the number of very long cells that were difficult to analyze. The analysis was restricted to cells of about double the size of a newborn cell, which likely originated from symmetric cell divisions and typically has two discernible nucleoids. To assay the extent of chromosome segregation in these cells, we stained the DNA with DAPI and determined the distribution of the DAPI signal along the long cell axis. Consistent with the involvement of MinD in chromosome segregation, we observed better separation of nucleoids in cells harboring wild-type MinD compared with those harboring the mutant MinD$^{R219D}$ (Figure 5C and D). This apparent difference could be confirmed by quantifying the distances between the centers of mass of the two nucleoids and also the depth of their separation (Figure 5E and F).

## MinD expression decreases mobility of chromosomal loci

According to our hypothesis, MinD molecules bind to the membrane and to the DNA and can tether the two, albeit only transiently. In agreement with that, we observed that expression of MinD in a Δ*minB* strain lowers the mobility of chromosomal loci associated with replication forks that were labeled with single-strand binding protein fused to YFP (SSB-YFP) (Possoz *et al*, 2006) (Figure 5G–I). Estimated apparent diffusion coefficients of these foci were ∼1.5 times lower in the presence of MinD (Supplementary Figure S6). Importantly, since the expression of MinD alone does not change the mini-cell phenotype typical of Δ*minB* cells, we could rule out that the observed difference in the mobility of replication forks was due to differences in cell morphology.

## Dynamics of the Min system can support proposed chromosome segregation

Taken together, our experimental results suggest that, in *E. coli* cells, the oscillating, membrane-bound MinD protein may indeed provide the source of the polar gradients of DNA tethering sites used for chromosome segregation. To confirm that the experimental parameters of the Min system are consistent with the proposed segregation mechanism, we

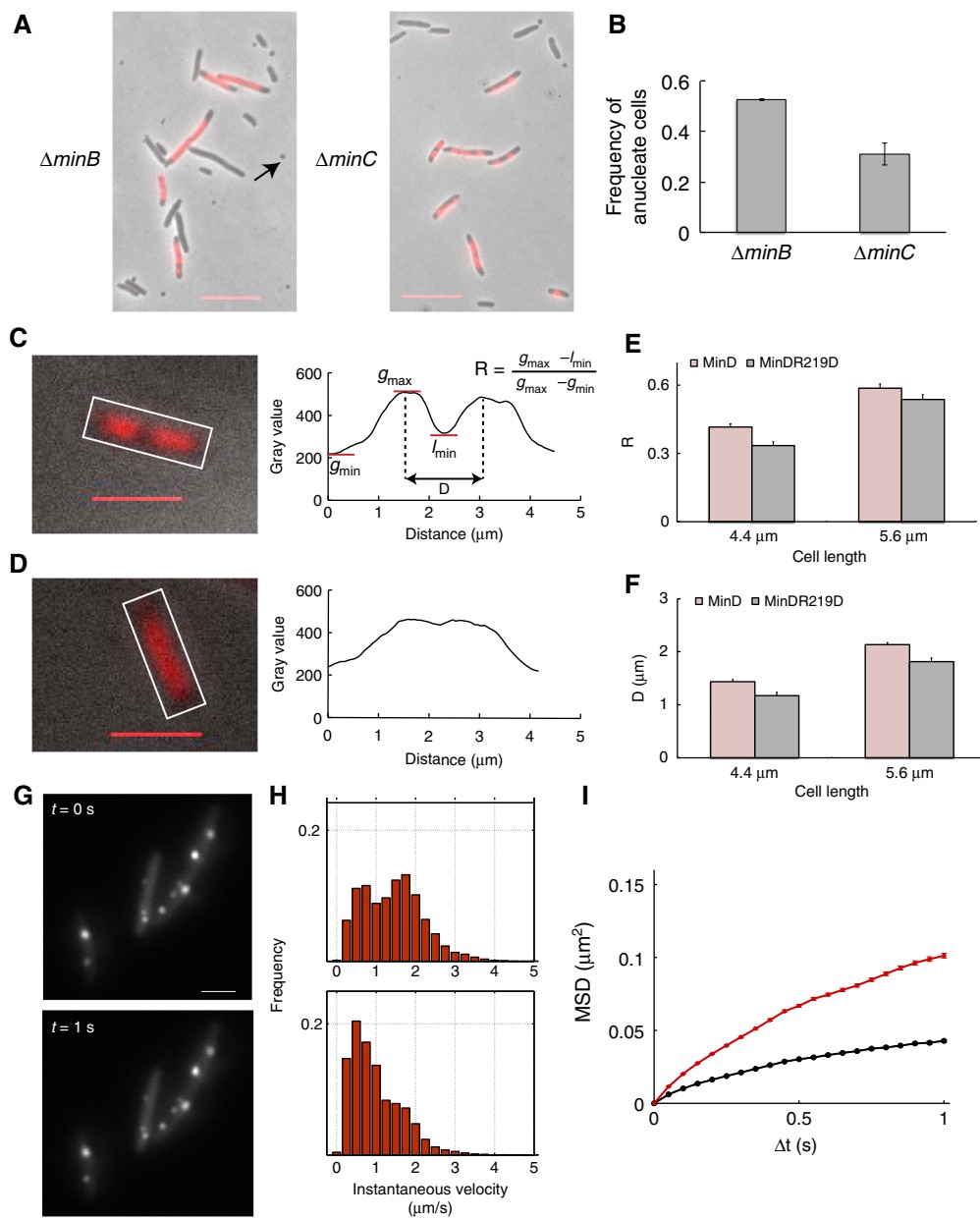

**Figure 5** MinD affects chromosome segregation and dynamics. (**A**) Representative images of Δ*minB* and Δ*minC* cells stained with DAPI to visualize the DNA. The overlay of the DAPI and phase-contrast images is shown. Cells were grown in rich medium at 37°C and samples were collected during exponential growth (OD$_{600}$ ∼0.2–0.5) and used for live-cell microscopy. The black arrowhead points to a mini-cell (mini-cells were not considered in the analysis). Scale bars, 5 μm. (**B**) Bar graph showing the number of cells without a DAPI signal divided by the total number of cells. The values represent the mean of two independent experiments, in which > 600 cells were analyzed. Error bars represent standard error of the mean. (**C–F**) Relative position of nucleoids in Δ*minB* strain expressing either wild-type (**C**) or mutant R219D (**D**) MinD along with MinE and MinC. Nucleoids were visualized using DAPI (red; left panels) and profiles of their intensity distribution along the long cell axis (right panels) were measured as indicated by the white box in the overlay of the fluorescence and the bright-field images. Scale bar, 3 μm. Individual intensity profiles in cells with two nucleoids were then evaluated for the depth (*R*) and the distance (*D*) of segregation, as illustrated. The mean values of *R* and *D* were plotted in (**E**) and (**F**), respectively, for cell sizes in the range of 3.78–4.97 μm (mean 4.4 μm) and in the range of 5.04–6.23 μm (mean 5.6 μm). (**G–I**) Mobility of replication forks labeled by SSB-YFP. Fluorescent foci of SSB-YFP were followed in a series of time-lapse microscopy images acquired every 50 ms. The magnitude of the frame-to-frame displacement was determined using automatic tracking for 10 920 and 10 141 individual foci in Δ*minB* cells expressing an empty plasmid or MinD at 10 μM for 4 h, respectively and SSB-YFP expression was induced with 0.01% arabinose for 4 h. (**G**) Two exemplary images of SSB-YFP foci (white arrows) in Δ*minB* cells. Scale bar, 2 μm. (**H**) Histograms of mean apparent velocity for individual foci in the absence (upper panel) or presence of MinD (lower panel). (**I**) Averaged curves for mean squared displacement (MSD) of SSB-YFP foci in the Δ*minB* cells in the presence (black) and absence (red) of MinD. Error bars represent standard error of the mean.

modified our simulations to more faithfully reflect the Min oscillations. For that, we first monitored spatial distribution of EYFP-MinD along the long cell axis in individual *E. coli* cells over time (Figure 6A; Supplementary Figure S7). We then used these measured profiles to describe the changes in the polar gradient of tethering sites in our simulations (Figure 6B). Despite differences in the details of gradient movement from the original simulation (Supplementary Figure S1C),

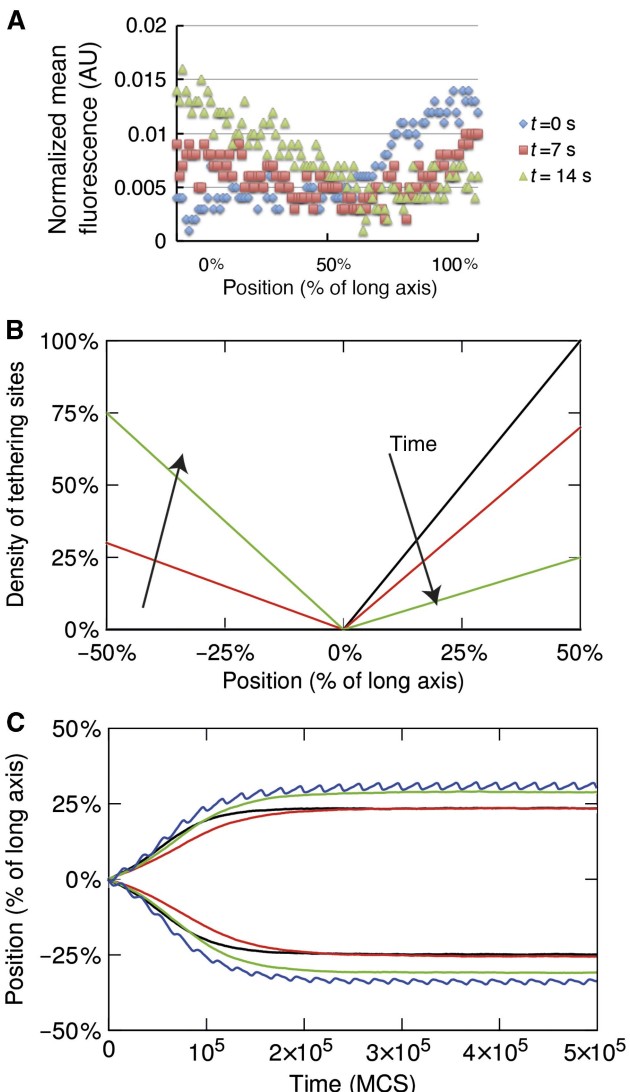

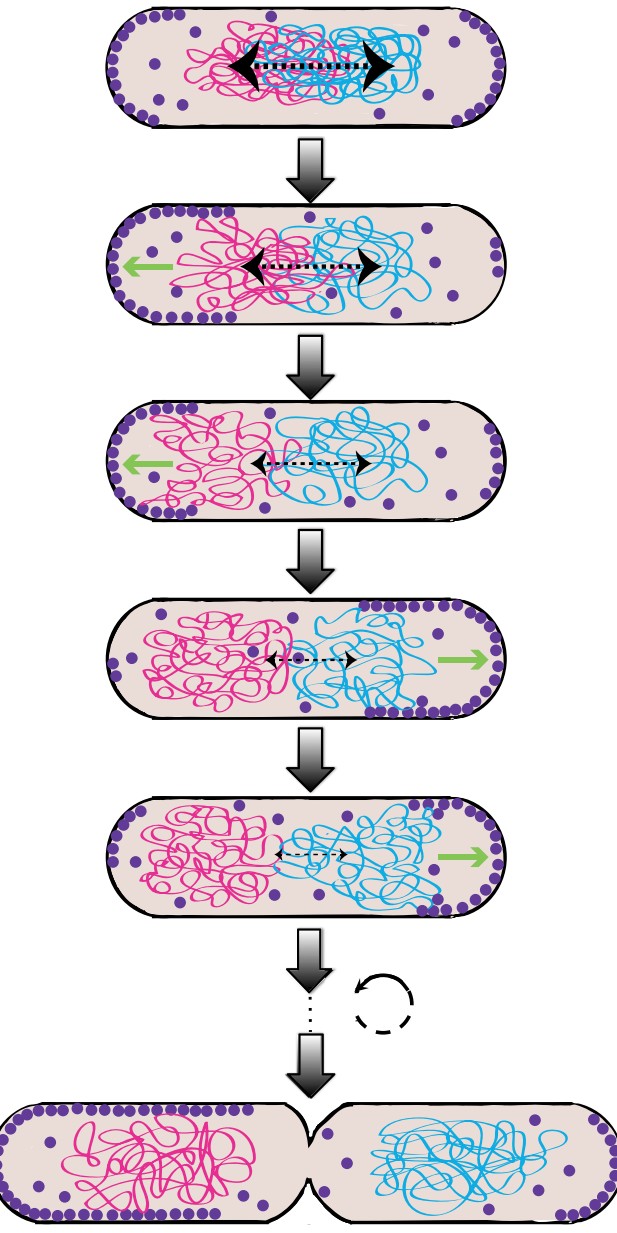

**Figure 6** Dynamics and numerical simulation of the Min system in chromosome segregation. (**A**) Spatial profiles of EYFP-MinD for selected time points of the Min oscillation cycle. EYFP-MinD was expressed in MG1655 cells at 30 μM IPTG induction and imaged for several minutes every 7 s. The mean fluorescence intensity profile was then plotted along the long cell axis at each time point for several oscillation cycles, with other time points shown in Supplementary Figure S6. (**B**) Schematics of the shape and movement of the MinD gradient based on (A). (**C**) Position of the center of mass (CM) of each polymer relative to the long cell axis as a function of time, measured in Monte-Carlo steps (MCS). All model parameters are the same as in Figure 1, expect that the oscillating gradient is implemented as in (B).

**Figure 7** Proposed model of chromosome segregation by the Min system. The cartoon shows a slowly dividing *E. coli* cell with only two duplicated chromosomes over time (from top to bottom). For simplicity, only MinD (violet dots) and only one DNA strand are shown. Entropic repulsive forces that move chromosomes apart are shown as black arrows pointing toward the poles, while the action of the Min system that creates an oscillating gradient of chromosome tethering sites is shown as green arrows pointing toward one pole during one cycle of the oscillation. See text for details.

simulations with such MinD-like gradient showed an almost identical improvement in chromosome segregation (Figure 6C). This indicates that it is primarily the existence of a gradient with a minimum at mid-cell and maxima at the poles and not the details of its movement that is critical for the proposed segregation mechanism.

## Discussion

Until now, the machinery used by *E. coli* and most other bacteria for the essential cellular function of chromosome segregation had remained elusive, with several previously proposed candidate processes being recently dismissed (Wang and Sherratt, 2010). In our model, we propose that non-sequence specific binding of MinD to DNA and at the same time to the membrane could create a dynamic gradient of DNA tethering sites on the membrane that progressively moves from mid-cell to the pole in each round of oscillation (Figure 7). Repeated binding and unbinding of chromosomal segments to these tethering sites eventually can mediate

segregation of sister chromosomes by biasing their random movement toward the poles in a Brownian ratchet-like manner. Our simulations demonstrate that the proposed mechanism can rely on either an oscillatory or a static gradient of tethering sites, meaning that may also function in bacterial species with static MinD localization (Marston *et al*, 1998). In the latter case of static gradient, the random movement of chromosomes is biased toward the poles at all times. In the former case of oscillatory gradient, each chromosome is biased in its movement toward the pole only during one half of the oscillation cycle, when MinD is mostly at the corresponding pole. During another half of the Min cycle, this chromosome is in principle freely diffusing, but because of the large size of the chromosome its diffusion is much slower than the cycle of Min oscillation (Jun and Mulder, 2006) and thus cannot randomize chromosome position on this time scale. Nevertheless, such alternating 'catch-and-release' can lead to oscillations of the center of mass of the chromosome above the gradual movement toward the pole, as observed in our simulations (Figure 1A). Notably, Fisher *et al* (2013) recently showed similar oscillations of the chromosome ('longitudinal density waves') on the time scale of tens of seconds, which we believe may be explained by the proposed Min-dependent chromosome tethering.

Importantly, to operate efficiently this mechanism requires the initial disentanglement of daughter nucleoids, which is likely achieved by entropic repulsion of self-avoiding ring chromosomes (Jun and Mulder, 2006) or by a related mechanism of minimization of radial confinement stress (Fisher *et al*, 2013). Consistent with previous reports (Jun and Mulder, 2006; Arnold and Jun, 2007; Jun and Wright, 2010; Bohn and Heermann, 2011), our work suggests that such entropic repulsion can efficiently push sister chromosomes apart during the early stages of segregation. However, because the entropic forces progressively weaken as the overlap between volumes occupied by the two chromosomes decreases, the entropic repulsion fails to achieve full segregation. Consistent with this analysis, daughter nucleoids show less efficient separation in the absence of the Min system but—in most cases—not a complete segregation defect. Notably, our simulations demonstrate that the proposed MinD-driven segregation should function independently of the details that underlie the initial unmixing of the chromosomes.

Given its well-established role in another essential process of the cell cycle, it is perhaps not surprising that the Min system was not considered as a likely candidate for chromosome segregation machinery, despite the homology of MinD to ParA and the early work demonstrating chromosome segregation defects in *min* strains (Akerlund *et al*, 1992, 2002). Consequently, previous comparisons between the ParA and the Min systems assumed that, despite similarities in their function and regulation, these systems have evolutionary diverged to execute two different key functions in bacterial cell division (Gerdes *et al*, 2010; Lutkenhaus, 2012). In contrast, our study proposes that the Min system in *E. coli* retained both functions, with the dynamic gradient of MinD on the membrane ensuring symmetric cell division and proper segregation of the daughter chromosomes. The interplay between these two functions apparently relies on an intimate intertwining of MinD interactions with the DNA and with the membrane. Different

from the DNA binding by ParA-type proteins, MinD interaction with DNA is further regulated by its membrane-binding amphipathic helix, although this sequence is not *per se* required for the DNA binding. On the other hand, mutations in the amino-acid residues that affect the DNA binding of MinD also apparently modulate its interaction with the membrane, although those residues are not part of the amphipathic helix.

The ATP dependence of both interactions is further likely to ensure that, in the cell, chromosomal DNA primarily interacts with membrane-bound MinD, thus reducing the non-productive sequestration of MinD at the nucleoid. Free diffusion of cytoplasmic MinD is, in fact, essential for the maintenance of the Min oscillations, which may also explain the observed relatively weak binding of MinD to DNA. Nevertheless, high local concentration of MinD at the membrane ensures that even its weak interactions with the chromosomal DNA are able to generate sufficient tethering force.

Proper partitioning of the genetic material is a key feature of the cell division process and it is controlled by multiple systems in bacteria (Reyes-Lamothe *et al*, 2012). It is therefore important to emphasize that during chromosome segregation, the Min machinery has to cooperate not only with the entropic forces but also with several other systems that have established roles in organizing the nucleoid throughout the cell cycle and in unlinking and translocating the concatenated daughter chromosomes through the closing septum (Wang *et al*, 2006; Danilova *et al*, 2007; Grainge *et al*, 2007; White *et al*, 2008; Madabhushi and Marians, 2009; Espeli *et al*, 2012; Reyes-Lamothe *et al*, 2012). Nevertheless, our work suggests that, similarly to eukaryotes, most bacteria employ a mitotic apparatus, although specific partitioning mechanisms in individual species might differ, relying either on the Min system or on the ParA system (Fogel and Waldor, 2006; Ptacin *et al*, 2010; Schofield *et al*, 2010; Shebelut *et al*, 2010).

## Materials and methods

### Strains and expression constructs

All strains and plasmids used in this study are listed in Supplementary Table S1. His-tagged MinD and its mutants were essentially purified as previously described (Loose *et al*, 2008). See Supplementary information for details of plasmid construction, cell growth conditions, and protein purification.

### DNA EMSAs

Binding reactions were performed in a volume of 10 µl in EMSA buffer (38 mM HEPES/NaOH (pH 7.2), 38 mM NaCl, 5 mM MgCl$_2$, 7% glycerol, 1 mM DTT). Each reaction contained 200 fmol of dsDNA labeled by 5′-hexachloro-fluorescein phosphoramidite (HEX) and 1 mM ATP or ADP (unless otherwise specified). Reactions were incubated at room temperature for 10 min and then separated on 10% polyacrylamide (PA) native gels for ∼30 min. Gels were run in 0.5X TBE plus 1 mM MgSO$_4$ at 150 V and subsequently visualized using a Typhoon gel scanner.

### Co-sedimentation and flotation assays

Co-sedimentation of MinD and DNA with liposomes is described in Supplementary information. Flotation assays were performed with unilamellar liposomes prepared from synthetic DOPG (1,2-dioleoyl-*sn*-glycero-3-phospho-(1′-*rac*-glycerol), sodium salt; Avanti Polar Lipids)

and 0.1% DiO (3,3′-dioctadecyloxacarbocyanine perchlorate; Invitrogen). Liposomes were incubated with HEX-labeled DNA and/or with recombinant wild-type or mutant MinD (40 µg/ml) and 1 mM ATP or ADP for 10 min at room temperature, and subsequently subjected to ultracentrifugation for 4 h at 48 000 r.p.m. and 4°C in the gradient of Nycodenz as described previously (Weber *et al*, 1998). All materials were recovered from the gradient in four fractions and fluorescence in respective fractions of the gradient was quantified using a Gemini XS plate reader (Molecular Devices). Additionally, MinD was quantified in each fraction using western blotting with anti-polyHistidine antibodies. See Supplementary information for details.

## Modeling and simulation of chromosome dynamics with and without the Min system

Two *E. coli* sister chromosomes were described as two self-avoiding ring polymers (Fritsche *et al*, 2012) that can move in an elongated rectangular parallelepiped of aspect ratio 1:8. For polymer rings of lengths $N = 80$, the linear dimensions of the confining geometry were set up such that the radius of gyration $R^{free}_{gyr}$ of the unconfined chain is larger than the linear square box sizes, leading to an $80 \times 10 \times 10$ lattice size and volume fraction of a single chain of 10%. Overlapping configurations of two chains, whose centers of mass coincide with the middle of the cell's long axis, were created to initiate the segregation process. Independent Monte-Carlo trajectories (different initial conditions driven by different random number sequences) representing the dynamics of the segregation process were then sampled.

Chromosome tethering was implemented by temporarily fixing monomers that approach the border of the confinement (see Supplementary information). Simulations were performed using the bond-fluctuation method (BFM) (Carmesin and Kremer, 1988), which has been applied successfully to model the static and dynamic properties of polymer systems in previous studies (Binder and Heermann, 2002).

## Supplementary information

## Acknowledgements

We thank Luis Serrano, Martin Howard, and Robert Grosse for comments on the manuscript, Lotte Sogaard-Andersen for support during a part of this work and members of the Lotte Sogaard-Andersen laboratory and Matthias Mayer for discussions. BDV and VS were funded by the CHS Foundation. BK and MF were financed by HGS MathComp. WJG was supported by the BMBF (FORSYS) project VIROQUANT. HA and WN were funded by the DFG SFB Transregio TRR83. The simulations were performed on bwGRiD (http://www.bw-grid.de).

*Author contributions:* BDV and VS conceived the study. BDV performed all experiments except flotation assays. WN and HA designed flotation assays. HA performed flotation assays. DWH, BK, and MF conceived chromosome dynamics simulations. MF wrote the code for the chromosome dynamics simulation. BK performed modeling and simulations. WJG wrote the tracking software and performed SSB velocity and MSD analyses under supervision of KR. All authors contributed to the discussion of results and participated in manuscript preparation. BDV and VS wrote the manuscript.

## Conflict of interest

The authors declare that they have no conflict of interest.

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
