## [Review Process File · Molecular Systems Biology]

Chromosome segregation by the Escherichia coli Min system

Barbara Di Ventura, Benoit Knecht, Helena Andreas, William J. Godinez, Miriam Fritsche, Karl Rohr, Walter Nickel, Dieter W. Heermann and Victor Sourjik

Corresponding author: Barbara Di Ventura, University of Heidelberg

Review timeline:

Submission date:	19 April 2013
Editorial Decision:	21 May 2013
Revision received:	26 June 2013
Editorial Decision:	02 August 2013
Revision received:	05 August 2013
Accepted:	07 August 2013

Editor: Maria Polychronidou

Transaction Report:

1st Editorial Decision

21 May 2013

Thank you again for submitting your work to Molecular Systems Biology. We have now heard back from the three referees who agreed to evaluate your manuscript. As you will see from the reports below, the referees raise substantial concerns on your work, which should be convincingly addressed in a revision of the manuscript.

Overall, the reviewers acknowledge that your work is addressing a potentially interesting topic. However, they point out that additional experimentation is required in order to convincingly support the proposed mechanism of chromosome segregation. Without repeating all the points listed by the reviewers, one of the more fundamental issues refers to the need to provide quantitative measurements of direct interactions and their effect on segregation dynamics. Reviewer #1 includes constructive suggestions in this regard. Moreover, many of the reviewers' comments refer to the need to better document several points throughout the manuscript and to provide additional controls.

If you feel you can satisfactorily deal with these points and those listed by the referees, you may wish to submit a revised version of your manuscript. Please attach a covering letter giving details of the way in which you have handled each of the points raised by the referees. A revised manuscript will be once again subject to

review and you probably understand that we can give you no guarantee at this stage that the eventual outcome will be favourable.

REFEREE REPORTS

Reviewer #1

In their paper, Di Ventura et al. propose the very appealing idea that the oscillating Min protein machinery, which is supposedly involved in the positioning of the divisome in *E. coli*, is also important for efficient chromosome segregation. Although disentanglement of mother and daughter nucleoid can be accomplished based on entropic forces only, it is much more efficient, particularly at later stages of separation, in the presence of an additional gradient force constituted by DNA binding proteins with higher concentration towards the cell poles than the middle. Since the Min machinery establishes such a (dynamic) gradient, and since MinD is a distant homologue to ParA shown to be involved in chromosome segregation in *C. crescentus*, it is a logical candidate to establish such a gradient force. Several experiments are presented that show the affinity of MinD to DNA. Although the amphipathic helix, or residues close to it, seem to play an important role in charge-mediated DNA binding, the membrane attachment of MinD seems to be not impaired by DNA binding.

Although the hypothesis is beautiful and appealing, and experimental evidence is presented that there is indeed weak binding between soluble and membrane-bound MinD and DNA, I find the results not yet convincing enough that the observed effect indeed contributes to a significant extent to the chromosome segregation *in vivo*. Additional, and in particular more quantitative, *in vitro* and/or *in vivo* measurements of direct interactions and their effect on segregation dynamics should be performed to render this study more significant.

What a convincing experimental-theoretical study should show is:

1. Time scales of spontaneous entropic segregation vs. Gradient-driven segregation (shown)
2. Dependence of segregation on steepness of gradient (not shown)
3. Dependence of segregation on association coefficients between DNA and MinD (not shown)
4. Measure steepness of gradient *in vivo* (not done)
5. Measure association coefficients *in vitro* (not done)

It is highly advisable to complement the missing information in this study to make it more coherent and quantitative.

Also, only MinD is promoted as the DNA-binding and segregation-enhancing protein. It is shown to bind DNA in an electrophoretic mobility assay. It is not surprising that a charged protein binds DNA. It should rather be shown that it binds DNA better or with higher affinity than any other protein that presumably forms a gradient, e.g., MinC. Also, a repulsive force (or a competing binding) by FtsZ or the machinery directing it to the cell poles could be envisaged.

Also, I find the interpretation of the centrifugation assays quite speculative. That MinD binds liposome membranes as monomers, and only in its multimeric binding is able to recruit DNA to the membrane is a result that would rather be demonstrated directly, e.g. in a fluorescence microscopy or spectroscopy assay, as easily accessible in any better facility these days.

The *in vivo* demonstration of complete vs. incomplete chromosome segregation seems to be ambiguous to me. I do not see how it can be inferred from the size of the daughter cells to what extent nucleoids have been segregated. There should be a better assay developed for that.

MinD lowers the mobility of chromosomal loci - is that proven to be related to the segregation efficiency? I would like to see a theoretical model for how mobility and segregation are coupled.

In conclusion, the punch line of this paper is beautiful, but the experimental evidence is not yet convincing enough. It should be carefully revised and resubmitted.

Reviewer #2

This is an interesting and potentially important paper. However, there are several ambiguities that make it difficult for the reader to be confident in the interpretations, and also some important questions that require answers before the work can be properly evaluated.

1. Gradient experiments-

- a. Why is there so much DNA in the top fraction in the absence of MinD? The text says "some" DNA, but this actually appears to be 50-75% of the total DNA. This is quite surprising and must be explained. If the DNA probe can actually bind phospholipid in the absence of MinD, might the MinD-induced change reflect DNA binding to phospholipid in vesicles that may run differently in the gradient because of MinD-induced changes in vesicle shape, size or configuration? This type of vesicle distortion has been reported to occur in previous MinD-vesicle binding studies.
- b. Is there evidence that these components are at their equilibrium positions in the gradients? If not, might the material in fraction 2 merely reflect smaller or misshapen vesicles that just have not yet moved to the top of the gradient?
- c. The graphs of the gradient experiments are somewhat unclear. The title in the upper portion of the graph (e.g., "HEX-DNA + MinD wt+DOPG") does not always seem to be congruent with the detailed information in the color-coding table below. Because of this it is not always completely clear what is measured on the vertical axis. The titles should be clarified. Also, I think it would be easier for the user if the vertical axis were labeled with the component being measured (i.e., DNA).
- d. It wasn't always clear in individual experiments whether or not the phospholipid was labeled with DiO, which I presume would be detected in the fluorescence measurements.
- e. Does the percent recovery shown on the vertical axis mean recovery as percent of

the total material recovered in the sum of the gradient fractions, or does it represent recovery as percent of the material applied to the gradient? If the former, it should be indicated how much of the applied material was actually recovered from the gradient. If this were a very low number it would cast some doubt on the interpretations.

2. Pelleting experiments-

I didn't see the essential control in which phospholipid was omitted.

3. Nucleoid segregation experiments-

a. If MinD actually plays a meaningful role in chromosome segregation I would expect an increase in anucleate cells (aside from the minicells). Was this observed?

b. In the SSP-Yfp mobility measurements, do the measurements distinguish the expected vectorial movement to the nearest pole from random directional movements?

4. Ratchet model-

I have some difficulty in seeing how this works since the poleward time-averaged gradients oscillate many times within a single division cycle. Does this imply that DNA domains are repeatedly released from the ratchet and are free to diffuse back toward midcell when MinD moves to the opposite end of the cell as part of each oscillation cycle? It would help the reader, I think, to explain how this would work at the molecular level.

It well may be that the authors can satisfactorily respond to these problems but, pending that outcome, the conclusions of the paper as written are not acceptable. Because of the potential significance of this work I hope the authors will be able to answer the questions raised above.

Reviewer #3

In this paper, the authors discover a new role for the Min system, previously implicated in cell division positioning in *E. coli*, also in chromosome segregation. The basic concept is that spatiotemporally oscillating MinD can bind non-specifically to DNA and to the membrane. In this way, MinD is able to act as a Brownian ratchet during chromosome segregation, thereby supplementing other division mechanisms such as entropic repulsion. The study is very nicely carried out, with a highly effective fusion of modelling and wet-lab approaches. I think it is a significant advance that will interest both microbiologists and systems biologists and is therefore appropriate for Molecular Systems Biology. I have a few comments that I hope the authors will find useful:

* Very recent work from Fisher et al Cell (2013) has shown that the bacterial chromosome is a highly organized object with a helix-like shape. Fisher et al claim that models in which the nucleoid is a randomly oriented polymer, defined by the cell cylinder, are excluded. I think the authors need to discuss these issues, as their simulations do employ an entropic repulsion component. Nevertheless, these new considerations cannot fundamentally alter the authors' conclusions, as they show in Fig 1E that simulated chromosome segregation proceeds normally even without entropic effects (where the role of MinD is even more prominent).

* Given this manuscript is partly theoretical in nature, I think it is appropriate to

cite the theoretical literature when discussing how the Min system works on p4 (e.g. Kruse et al Mol Microbiol 2007).

* On p4, the authors propose that "a gradient of DNA binding sites at the cell membrane can pull chromosomes...". I'm not sure this is quite right. MinD simply tethers DNA to the membrane, preventing backward movement of DNA towards midcell. The wording here and in a few other places therefore needs improving.

* In the simulations of Figure 1, is there any way of relating the results back to real time rather than Monte-Carlo steps?

* There is a typo in the axis labelling of Fig 5G.

1st Revision - authors' response

26 June 2013

Reviewer #1

In their paper, Di Ventura et al. propose the very appealing idea that the oscillating Min protein machinery, which is supposedly involved in the positioning of the divisome in *E. coli*, is also important for efficient

chromosome segregation. Although disentanglement of mother and daughter nucleoid can be accomplished based on entropic forces only, it is much more efficient, particularly at later stages of separation, in the presence of an additional gradient force constituted by DNA binding proteins with higher concentration towards the cell poles than the middle. Since the Min machinery establishes such a (dynamic) gradient, and since MinD is a distant homologue to ParA shown to be involved in chromosome segregation in *C. crescentus*, it is a logical candidate to establish such a gradient force. Several experiments are presented that show the affinity of MinD to DNA. Although the amphipathic helix, or residues close to it, seem to play an important role in charge-mediated DNA

binding, the membrane attachment of MinD seems to be not impaired by DNA binding.

Although the hypothesis is beautiful and appealing, and experimental evidence is presented that there is indeed weak binding between soluble and membrane-bound MinD and DNA, I find the results not yet convincing enough that the observed effect indeed contributes to a significant extent to the chromosome segregation *in vivo*. Additional, and in particular more quantitative, *in vitro* and/or *in vivo* measurements of direct interactions and their effect on segregation dynamics should be performed to render this study more significant.

What a convincing experimental-theoretical study should show is:

1. Time scales of spontaneous entropic segregation vs. Gradient-driven segregation (shown)
2. Dependence of segregation on steepness of gradient (not shown)
3. Dependence of segregation on association coefficients between DNA and MinD (not shown)
4. Measure steepness of gradient *in vivo* (not done)
5. Measure association coefficients *in vitro* (not done)

It is highly advisable to complement the missing information in this study to make it more coherent and quantitative.

First of all, we thank the reviewer for recognizing the beauty and appeal of our proposed mechanism and for suggesting further important experiments and computer simulations to strengthen the proposed model of chromosome segregation.

On the experimental side (points 4 and 5), we performed surface plasmon resonance (Biacore) studies to estimate the rates of MinD binding to DNA (Figure 2F). We found that the rates of binding and dissociation are comparable to the rates observed for other DNA-binding proteins. Moreover, the measured time scales of MinD-DNA interaction are similar to the time scale of the Min oscillation, which sets the limit of the dwell time of the chromosome at the membrane in our simulations. Furthermore, we have now quantified the steepness of the MinD gradient *in vivo* at different time points of the Min

oscillation (Supplementary Figure 6 and Figure 6A).

On the theoretical side (points 2 and 3) we have now carried out simulations with a shallower gradient (Supplementary Figure 1D) and with a 10 times stronger affinity of the DNA for the tethering sites (Supplementary Figure 1E). These simulations show that a shallower gradient still works but is much less efficient than the steeper one, and that a stronger affinity for the tethering sites impairs segregation, consistent with our hypothesis that MinD-DNA binding ought to be weak. We now better explain the details of the simulation in the text and in Supplementary Information (page 6, first paragraph; pages 11-14).

Finally, we have performed new simulations with a gradient whose movement more exactly represents the one determined in cells (new Figure 6).

Also, only MinD is promoted as the DNA-binding and segregation-enhancing protein. It is shown to bind DNA in an electrophoretic mobility assay. It is not surprising that a charged protein binds DNA. It should rather be shown that it binds DNA better or with higher affinity than any other protein that presumably forms a gradient, e.g., MinC. Also, a repulsive force (or a competing binding) by FtsZ or the machinery directing it to the cell poles could be envisaged.

While we agree with the reviewer that in principle positively charged proteins might show a weak non-specific DNA binding, we believe that

such binding is unlikely under stringent conditions used in our assays. Moreover, MinD overall is not positively charged. Its estimated pI of 5.37 suggests that at physiological pH levels the overall charge of MinD may be negative rather than positive. Although the C-terminal helix of MinD that interacts with the cytoplasmic membrane is indeed positively charged, we show in Figure 2E that MinD^{R2E,10} mutant that lacks this helix but carries two arginine-to-glutamate substitutions (which adds two further negatively charged residues in place of two positive ones) can bind the DNA *in vitro* similar to the wild-type MinD. This strongly argues against the overall protein charge (or the charge of the membrane-binding helix) as the determinant of DNA binding. We further observe that the binding of MinD to DNA *in vitro* is abolished by several specific mutations, which in our opinion provides sufficient control for the specificity of the interaction. And apparent involvement of at least one of the ParA-conserved residues (R219) suggests that the binding mode of MinD and ParA to DNA might be similar, supporting our original hypothesis.

Moreover, we now include a new result (new Figure 4E) showing that MinD^{R2E,10}-YFP, which binds to the DNA but not to the membrane *in vitro*, is indeed enriched on the nucleoid *in vivo*. Together with the observed effect of MinD on the intracellular mobility of SSB-EYFP loci in absence of other Min proteins (Figure 5 E-G), we believe that this observation presents sufficient evidence that MinD does bind to the DNA specifically and tethers it to the membrane *in vivo*. It is important to emphasize that - as confirmed by our simulations - the proposed

segregation mechanism requires the binding of MinD to chromosomal DNA to be relatively weak and transient, as stronger binding would impair Min oscillation. This also explains why chromosomal localization cannot be observed for the wild-type MinD, which is found predominantly on the cytoplasmic membrane.

Also, I find the interpretation of the centrifugation assays quite speculative. That MinD binds liposome membranes as monomers, and only in its multimeric binding is able to recruit DNA to the membrane is a result that would rather be demonstrated directly, e.g. in a fluorescence microscopy or spectroscopy assay, as easily accessible in any better facility these days.

While it is principally also possible to study MinD-DNA-liposomes association with other techniques, we believe that an *in vitro* reconstitution of the system with purified components - as performed in our flotation assay - is excellently suited to show such interactions. The flotation assay is well established and much used in the Nickel's group, and in our opinion it provides direct evidence of the MinD-mediated tethering of the DNA to the membrane.

We have now modified and simplified Figure 3 to clarify the interpretation of these experiments. In a nutshell, the assay shows that DNA is specifically enriched in the top fractions 1 and 2 in the presence of liposomes and MinD. While the enrichment in the fraction

1 may be at least partly mediated by a non-specific association of DNA with liposomes themselves, the recruitment of liposomes and DNA to fraction 2 is clearly MinD- and ATP-dependent, suggesting that this fraction contains membrane-MinD*ATP-DNA complexes. We further show now that the ratio of DNA to liposomes is higher in fraction 2, confirming that recruitment of DNA to fraction 2 cannot be expected by the non-specific binding to liposomes. Moreover, higher ratio of DNA to MinD in fraction 2 strongly suggests that the membrane-bound multimeric MinD*ATP present in this fraction is indeed more efficient in DNA recruitment than MinD present in fraction 1. Both the text and the figure have now been modified to emphasize these points.

Moreover, these *in vitro* analyses are further supported by *in vivo* evidence of the binding of MinD to DNA (Figure 4E) and of MinD-mediated DNA tethering to the membrane (see below).

The *in vivo* demonstration of complete vs. incomplete chromosome segregation seems to be ambiguous to me. I do not see how it can be inferred from the size of the daughter cells to what extent nucleoids have been segregated. There should be a better assay developed for that.

We apologize for the confusion that must have arisen from our wording. We do not infer the chromosome segregation from the size of the daughter cells. Rather, we used cells of the same size (approximately corresponding to double the size of a newly divided

cell) but expressing either wild-type MinD or MinD^{R219D} and directly compared the extent of the segregation of nucleoids stained with DAPI. Similar analysis has been done previously (Akerlund et al, 2002). Separation of nucleoids is assayed by plotting the fluorescence profile along the long axis of the cell. If the chromosomes are well separated, the profile of DAPI fluorescence along the long cell axis will show two discernable maxima, with a trough in the middle (Figure 5A). If the segregation is not complete, the profile will be more continuous, indicating that the DAPI signal is present also at mid-cell (Figure 5B). We now changed the text to explain this point better (page 13).

MinD lowers the mobility of chromosomal loci - is that proven to be related to the segregation efficiency? I would like to see a theoretical model for how mobility and segregation are coupled.

We apologize for the apparent ambiguity. The goal of the experiment referred to by the reviewer (Figure 5 E-G) was only to confirm that MinD can bind DNA *in vivo* and can indeed tether it to the membrane. It is done in a different background (*min*⁻) than the chromosome segregation analysis, because in the absence of other Min proteins we expect the MinD-dependent tethering on DNA to the membrane to be more apparent, due to more stable uniform localization of MinD to the membrane.

While such uniform tethering (and thus reduction of mobility) by itself cannot drive segregation, tethering along a polar gradient of binding

sites can. These scenarios (uniform tethering, static and oscillatory polar gradients) are indeed explored in our simulations (Figure 1).

In conclusion, the punch line of this paper is beautiful, but the experimental evidence is not yet convincing enough. It should be carefully revised and resubmitted.

We hope that our additional experiments together with better wording and explanation of the proposed segregation mechanism convinced the reviewer that our data do support our model of Min-mediated chromosome segregation.

Reviewer #2 (Remarks to the Author):

This is an interesting and potentially important paper. However, there are several ambiguities that make it difficult for the reader to be confident in the interpretations, and also some important questions that require answers before the work can be properly evaluated.

We thank the reviewer for finding our data potentially interesting and important for the scientific community. We hope that our additional experiments and clarification of our conclusions provide sufficient support to our model.

1. Gradient experiments-

a. Why is there so much DNA in the top fraction in the absence of MinD? The text says "some" DNA, but this actually appears to be 50-75% of the total DNA. This is quite surprising and must be explained. If the DNA probe can actually bind phospholipid in the absence of MinD, might the MinD-induced change reflect DNA binding to phospholipid in vesicles that may run differently in the gradient because of MinD-induced changes in vesicle shape, size or configuration? This type of vesicle distortion has been reported to occur in previous MinD-vesicle binding studies.

We do agree with the reviewer that it is important to ensure that what we observe is indeed a MinD-mediated tethering of DNA to the liposomes and does not result from an unspecific association of the DNA with the liposomes. We would like first of all to point out that the percentage of DNA floating to fraction 1 together with liposomes is less than 25% (Supplementary Figure 3C). Moreover, the percentage of liposomes in fraction 1 is in this case (only DNA+liposomes) almost 80% of the total. When MinD is added to the reaction, we get about 20% of the total liposomes going to fraction 2, but the percentage of DNA is much higher, clearly suggesting that the tethering of DNA to liposomes in fraction 2 cannot be explained by an unspecific binding. To make this point clearer we now revised Figure 3 to include only the most relevant information, leaving the details of the experiment to the

supplement. We further added to the main figure graphs representing the ratio between DNA and liposomes as well as the ratio between DNA and protein, to support our conclusion that the enrichment of DNA in fraction 2 is specifically mediated by MinD*ATP. We also modified the text accordingly (pages 10-11).

b. Is there evidence that these components are at their equilibrium positions in the gradients? If not, might the material in fraction 2 merely reflect smaller or misshapen vesicles that just have not yet moved to the top of the gradient?

The flotation assay is well established and has been optimized (Steringer et al, 2012; Weber et al, 1998) to ensure that the components have reached their equilibrium after 4h of centrifugation. Thus, we are confident that the material is in equilibrium and does not reflect material that would still float to another fraction. We comment on this in the revised version of the manuscript (page 10).

c. The graphs of the gradient experiments are somewhat unclear. The title in the upper portion of the graph (e.g., "HEX-DNA + MinD wt+DOPG") does not always seem to be congruent with the detailed information in the color-coding table below. Because of this it is not always completely clear what is measured on the vertical axis. The titles should be clarified. Also, I think it would be easier for the user if the vertical axis were labeled with the component being measured (i.e.,

DNA).

We thank the reviewer for pointing out the ambiguity in the data presentation. We have now substantially changed the main figure (see new Figure 3) to make it simpler and more to the point and have decided to show each individual component separately in the supplement (Supplementary Figure 3). We also re-labeled the axes to make clear that what is measured is the percentage of indicated material (DNA, DOPG or MinD) recovered in each fraction compared to the total material recovered from the gradient.

d. It wasn't always clear in individual experiments whether or not the phospholipid was labeled with DiO, which I presume would be detected in the fluorescence measurements.

In all flotation experiments the liposomes were labeled with DiO and detected via fluorescence in the plate reader, and this is what is plotted as DOPG fraction in the figure. In the sedimentation experiment showed in Figure 4 and Supplementary Figure 4, we used un-labeled liposomes.

e. Does the percent recovery shown on the vertical axis mean recovery as percent of the total material recovered in the sum of the gradient fractions, or does it represent recovery as percent of the material applied to the gradient? If the former, it should be indicated how

much of the applied material was actually recovered from the gradient. If this were a very low number it would cast some doubt on the interpretations.

We recover and measure all of the material in the four fractions, thus recovering the majority of the total material applied to the gradient. We now clarify that the Y-axis shows the amount of material as a percentage of the total material recovered from the gradient.

2. Pelleting experiments-

I didn't see the essential control in which phospholipid was omitted.

We now show additional pelleting experiments in which this control was included (new Supplementary Figure 4)

3. Nucleoid segregation experiments-

a. If MinD actually plays a meaningful role in chromosome segregation

I would expect an increase in anucleate cells (aside from the minicells).

Was this observed?

As DAPI does not stain all cells, it is first of all difficult to ascertain if a non-mini-cell is DNA-free because of absence of DNA or because of a failure in DAPI staining. Moreover, given that anucleate cells are rarely found even in MukB (SMC) deficient strains where chromosome organization is strongly perturbed, we do not expect to see a major increase in anucleate cells in min^- strains. Incomplete or delayed

chromosome segregation in min^- cells it likely to inhibit or delay division in a fraction of cells due to the nucleoid occlusion but would not necessarily lead to anucleate cells that are not mini-cells.

b. In the SSP-Yfp mobility measurements, do the measurements distinguish the expected vectorial movement to the nearest pole from random directional movements?

These experiments were performed to verify our prediction that MinD binding to DNA and membrane *in vivo* should tether chromosomal segments to the membrane and reduce their mobility. This prediction was indeed confirmed. In the strain used for this experiment, however, there is no oscillation of the Min system, since MinD is expressed in the absence of the other Min proteins. Prompted by the reviewer's question, we now performed preliminary analysis of SSB-YFP mobility in the wild-type strain compared to min^- strain. These data indeed suggest a more directed movement of SSB-YFP loci in the wild-type background, but the effect was modest and confirming and exploring it would require a much larger effort that would go beyond the scope of this manuscript and will thus be done as a separate study.

4. Ratchet model-

I have some difficulty in seeing how this works since the poleward

time-averaged gradients oscillate many times within a single division cycle. Does this imply that DNA domains are repeatedly released from the ratchet and are free to diffuse back toward midcell when MinD moves to the opposite end of the cell as part of each oscillation cycle? It would help the reader, I think, to explain how this would work at the molecular level.

We do agree with the reviewer that it is essential to explain well in the paper how chromosome segregation by the Min system works at the molecular level. The cartoon in (now) Figure 7 was intended for this purpose, but perhaps it failed to capture all the relevant information because we tried to keep it minimal to avoid a too crowded representation. The reviewer is right: in each oscillation cycle, the sister chromosomes are alternatively exposed to the tethering only when the MinD polar zone is there. Our model indeed relies on the assumption that diffusional movement of the chromosome is substantially slower than the cycle of the Min oscillation, so that the chromosome does not have the time to entirely randomize its position during the half of the Min cycle that it takes for MinD to get back to that pole. We believe that this assumption is well justified by the large size of the chromosome. Indeed, the apparent diffusion of chromosomal segments packaged within the nucleoid has been estimated to be $\sim 10^{-5} \mu\text{m}^2 \text{sec}^{-1}$, implying the characteristic time of 10^5 sec for diffusion on $1 \mu\text{m}$ scale (Jun & Mulder, 2006), which is much slower than the cycle of Min oscillation. We now explained and

justified this assumption more explicitly in the text (pages 15-16, last sentence and first sentence, respectively).

Notably, in our simulations, we actually do see small “oscillations” of the polymer’s centre of mass due to the backwards movement, which does not prevent the ultimate the segregation (see Figure 1A).

It well may be that the authors can satisfactorily respond to these problems but, pending that outcome, the conclusions of the paper as written are not acceptable. Because of the potential significance of this work I hope the authors will be able to answer the questions raised above.

Reviewer #3 (Remarks to the Author):

In this paper, the authors discover a new role for the Min system, previously implicated in cell division positioning in *E. coli*, also in chromosome segregation. The basic concept is that spatiotemporally oscillating MinD can bind non-specifically to DNA and to the membrane. In this way, MinD is able to act as a Brownian ratchet during chromosome segregation, thereby supplementing other division mechanisms such as entropic repulsion. The study is very nicely carried out, with a highly effective fusion of modelling and wet-lab approaches. I think it is a significant advance that will interest both

microbiologists and systems biologists and is therefore appropriate for *Molecular Systems Biology*.

We thank the reviewer for considering our work appropriate for publication in *Molecular Systems Biology* and for appreciating that our study is nicely carried out and shows a good combination of experiments and theory.

I have a few comments that I hope the authors will find useful:

* Very recent work from Fisher et al Cell (2013) has shown that the bacterial chromosome is a highly organized object with a helix-like shape. Fisher et al claim that models in which the nucleoid is a randomly oriented polymer, defined by the cell cylinder, are excluded. I think the authors need to discuss these issues, as their simulations do employ an entropic repulsion component. Nevertheless, these new considerations cannot fundamentally alter the authors' conclusions, as they show in Fig 1E that simulated chromosome segregation proceeds normally even without entropic effects (where the role of MinD is even more prominent).

As the reviewer points out, we show that the proposed Min-mediated segregation functions independently of the mechanism underlying the initial unmixing of the chromosomes, be it the entropic repulsion or the radial-confinement stress as proposed by Fisher *et al.* (which is

ultimately a kind of entropic repulsion). We now stress this fact in the text (page 16, second paragraph). Moreover, Fischer *et al.* make a very interesting observation of periodic longitudinal waves of chromosome density. The origin of these waves could not be explained by the authors but we believe that DNA tethering by the oscillatory Min system can perfectly account for this observation.

* Given this manuscript is partly theoretical in nature, I think it is appropriate to cite the theoretical literature when discussing how the Min system works on p4 (e.g. Kruse et al Mol Microbiol 2007).

We agree with the reviewer and have added this citation.

* On p4, the authors propose that "a gradient of DNA binding sites at the cell membrane can pull chromosomes...". I'm not sure this is quite right. MinD simply tethers DNA to the membrane, preventing backward movement of DNA towards midcell. The wording here and in a few other places therefore needs improving.

We thank the reviewer for this criticism. We have now changed the wording accordingly to the proposed Brownian-ratchet model and we do not speak anymore of a "pulling" force.

* In the simulations of Figure 1, is there any way of relating the results back to real time rather than Monte-Carlo steps?

While it is in principle possible to roughly estimate a relationship between Monte-Carlo steps and real time by comparing the measured diffusion constant of DNA inside the cell with the diffusion constant of simulated DNA chains, the resulting relationship is only an indication of the order of magnitude and should not be considered as a real conversion parameter, as many of the values involved need to be guessed.

Having said that, based on (Jun & Mulder, 2006), the experimentally measured diffusion constant for a chromosomal focus within the nucleoid is $\sim 10^{-5} \mu\text{m}^2 \text{sec}^{-1}$. The diffusion constant in our simulations is $10^{-3} \text{lu}^2 \text{MCS}^{-1}$ (where lu is lattice units). Assuming that cells are about $1 \mu\text{m}$ in size - which corresponds to 80 lu in our simulations - we get $1 \text{sec} \sim 640 \text{MCS}$, meaning that the segregation takes place in about 300 seconds ($2 \times 10^5 \text{MCS}$). For comparison, Min oscillation in the model takes $2 \times 10^4 \text{MCS}$. Using the estimate above, this would correspond to 30 sec, which is indeed similar to the experimentally observed period.

Given that the conversion between MCS and real time is only a very rough estimation, we prefer not to comment explicitly on this in the text.

* There is a typo in the axis labelling of Fig 5G.

We thank the reviewer for pointing this out to us. We have corrected the typo.

Akerlund T, Gullbrand B, Nordstrom K (2002) Effects of the Min system on nucleoid segregation in Escherichia coli. *Microbiology* **148**: 3213-3222

Elowitz MB, Surette MG, Wolf PE, Stock JB, Leibler S (1999) Protein mobility in the cytoplasm of Escherichia coli. *Journal of bacteriology* **181**: 197-203

Jun S, Mulder B (2006) Entropy-driven spatial organization of highly confined polymers: lessons for the bacterial chromosome. *Proc Natl Acad Sci U S A* **103**: 12388-12393

Steringer JP, Bleicken S, Andreas H, Zacherl S, Laussmann M, Temmerman K, Contreras FX, Bharat TA, Lechner J, Muller HM, Briggs JA, Garcia-Saez AJ, Nickel W (2012) Phosphatidylinositol 4,5-bisphosphate (PI(4,5)P₂)-dependent oligomerization of fibroblast growth factor 2 (FGF2) triggers the formation of a lipidic membrane pore implicated in unconventional secretion. *The Journal of biological chemistry* **287**: 27659-27669

Weber T, Zemelman BV, McNew JA, Westermann B, Gmachl M, Parlati F, Sollner TH, Rothman JE (1998) SNAREpins: minimal machinery for membrane fusion. *Cell* **92**: 759-772

Thank you again for submitting your work to Molecular Systems Biology. We have now heard back from the three referees who agreed to evaluate your manuscript. As you will see from the reports below, while the main concerns of reviewers #1 and #3 have been satisfactorily addressed, reviewer #2 points out that there are still two issues that need to be dealt with. These issues regard i) the floatation assays and ii) the presence of anucleate cells in the absence of MinD. Additional experimentation regarding the floatation assay would not be required. (Of course, if such experiments have already been performed we would welcome their inclusion.) However, we would like to ask you to provide some additional clarifications/explanations concerning the second point of reviewer #2, regarding the segregation defects in MinD mutant cells.

REFEREE REPORTS

Reviewer #1

The authors have dealt with the raised criticism in an adequate and thoughtful manner. Clearly, there are several questions still open, but the novelty of the concept and originality of hypotheses definitely warrant publication at this point. This is without doubt a stimulating paper which will inspire the community. It should be published as is.

Reviewer #2

The authors have responded to the previous comments and the revised manuscript is significantly improved. However, there are two points which are still problematic. The first especially is so important that I do not believe the manuscript is acceptable until the necessary experiments have been done.

1. Were the samples at their true equilibrium positions in the density gradients? The authors say that this has been well-established and need not be reconfirmed. I disagree. The important gradient fractionation experiments absolutely depend on the samples having reached their buoyant density positions. Firstly, the papers quoted by the authors concerned lipid vesicles prepared with completely different lipids, and the ratios of DNA, proteins and lipids in the present experiments almost surely altered the buoyant density of the vesicles. This alone would require independent demonstration that equilibrium had been reached. Second, the rate of movement of particles in density gradients is significantly affected by the size of the particles (vesicles) and their overall conformation. Unilamellar vesicles prepared in different laboratories, especially with different lipids, may vary significantly in size. In addition, deformed vesicles or fused vesicles induced by the additional components can have a real effect on sedimentation velocity and hence on the time required for the vesicles to reach their equilibrium positions in the gradient. In any case, the required experiments are simple. I believe for the few key experiments the tubes should be spun for two time periods, four hours to correspond to the bulk of

the experiments, and a longer period (?8 hours) to prove that equilibrium has been achieved. Once this has been established the point will have been addressed. The absence of this information is, in my opinion, a fatal flaw.

2. Why are there apparently no anucleate cells in the absence of MinD? The authors say that: 1. DAPI doesn't stain all cells even in wild type situations, making it difficult to determine a small increase in anucleate cells. In fact, in most studies of DAPI-labeled bacterial cells almost all cells contain visible nucleoid fluorescence so it really shouldn't be difficult to do these important measurements. 2. They suggest that even in known chromosome segregation mutants, such as MukB mutants, there are few anucleate cells. I do not believe this is true, either for MukB (where the original identification of the mutants was based on the formation of anucleate cells) and for other segregation-defective mutants in which anucleate cells were easily detected by DAPI staining. 3. The authors say that in the absence of MinD there may be a slowing of chromosome segregation that is not sufficient to produce anucleate cells. This is, or course, possible. However, I think the main interest of this paper is the idea that MinD plays a significant role in normal segregation, in which the whole point is to ensure that each daughter cell receives one of the two daughter chromosomes. If the effect is so marginal that there is neither anucleate cell production nor guillotining of the predicted poorly segregated terminus regions of the chromosome, then one wonders whether the claimed phenomenon has any biological significance. I like their model but believe this is a troublesome aspect that needs more consideration than it has received. It would be well worth the effort, in my opinion, to do some experiments to ask whether there is a detectable effect on equitransmission of chromosomes to the daughter cells or an effect of the integrity of chromosomes, which presumably be manifested also by formation of numbers of nonviable cells because of loss of chromosomal domains in the terminus region.

I do not believe this paper is acceptable until these issues have been dealt with.

Reviewer #3

The authors have satisfactorily addressed my concerns and I am now happy to see the ms published in MSB.

2nd Revision - authors' response

05 August 2013

We were very pleased to read that all reviewers found our revised manuscript largely improved and that at least reviewers #1 and #3 thought that the manuscript is ready now for publication in *Molecular Systems Biology*.

As for the remaining concern of Reviewer #2 regarding the role of MinD in chromosome segregation *in vivo*, we have added a new data set (new panels A and B in Figure 5 and new Supplementary Figure 5) showing the results of an experiment with DAPI staining we performed meanwhile.

Albeit we still believe that DAPI staining is not very accurate to determine the absolute number of anucleate cells (beyond the mini-cells),

given the fact that this staining is not necessarily efficient in all cells, we decided to test the reviewer's hypothesis that absence of MinD should correspond to an increase in anucleate cells nonetheless. We therefore compared the frequency of cells lacking the DAPI signal in strains deleted for the entire *minB* operon ($\Delta minB$) or only for *minC* ($\Delta minC$). We indeed detected an increase in anucleate cells in the former strain, a result that supports the proposed role of MinD in chromosome segregation *in vivo*. This experiment also shows that chromosome morphology and separation is much more defective in the $\Delta minB$ strain as compared to the $\Delta minC$ strain, indicating that the presence of a MinD gradient (in the $\Delta minC$ strain, in which MinD and MinE still oscillate) is helping chromosome segregation to a detectable extent.

Finally, we would like to comment on the criticism of reviewer #2 regarding the flotation assay. The reviewer is probably right in that we cannot directly compare our experiments to those in the papers we cited since we used indeed other lipid compositions and proteins. However, while these factors may influence the time needed to reach the equilibrium, the effects are likely to be small. We believe that it is extremely unlikely that the type of Nycodenz gradient (a low viscosity material) we are using (4 hours at 150,000 g) does not result in an equilibrium distribution of the membranes corresponding to their buoyant density. This is a standard gradient that has been designed to separate a broad range of membrane vesicles based upon equilibrium distribution according to their buoyant density. Even velocity-controlled gradients using materials with higher viscosity (such as sucrose) are typically centrifuged no longer than 15 min. Taken together all these considerations, we are confident that centrifuging the samples for 8 hours would not lead to significantly different results. Moreover, the *in vitro* flotation assays used to show that MinD tethers DNA to the membrane are nicely reflected in the *in vivo* SSB mean square displacement analysis we performed.